# Computational Mechanisms Mediating Inhibitory Control of Coordinated Eye-Hand Movements

**DOI:** 10.3390/brainsci11050607

**Published:** 2021-05-10

**Authors:** Sumitash Jana, Atul Gopal, Aditya Murthy

**Affiliations:** 1Department of Psychology, University of California San Diego, La Jolla, CA 92093, USA; 2Laboratory of Sensorimotor Research, National Eye Institute, Bethesda, MD 20814, USA; 3Centre for Neuroscience, Indian Institute of Science, Bangalore, Karnataka 560012, India; adi@iisc.ac.in

**Keywords:** accumulator model, race model, flexible behavior, reaction time variability

## Abstract

Significant progress has been made in understanding the computational and neural mechanisms that mediate eye and hand movements made in isolation. However, less is known about the mechanisms that control these movements when they are coordinated. Here, we outline our computational approaches using accumulation-to-threshold and race-to-threshold models to elucidate the mechanisms that initiate and inhibit these movements. We suggest that, depending on the behavioral context, the initiation and inhibition of coordinated eye-hand movements can operate in two modes—coupled and decoupled. The coupled mode operates when the task context requires a tight coupling between the effectors; a common command initiates both effectors, and a unitary inhibitory process is responsible for stopping them. Conversely, the decoupled mode operates when the task context demands weaker coupling between the effectors; separate commands initiate the eye and hand, and separate inhibitory processes are responsible for stopping them. We hypothesize that the higher-order control processes assess the behavioral context and choose the most appropriate mode. This computational mechanism can explain the heterogeneous results observed across many studies that have investigated the control of coordinated eye-hand movements and may also serve as a general framework to understand the control of complex multi-effector movements.

## 1. Introduction

To lead productive lives, we need to rapidly stop contextually inappropriate movements, such as stepping onto the street when we see a car approaching or speaking when a lecture starts. Such forms of inhibitory control, i.e., executive control processes that mediate behavioral action-stopping, have usually been studied using simple movements, such as eye movements and button presses, reviewed in [1,2]. However, the majority of our daily activities involve complex multi-effector movements. Scant attention has been paid to studying the neural and computational mechanisms that mediate the stopping of such complex movements. In this review, we summarize our series of studies into one such complex two-effector system—coordinated eye-hand movements.

Many of our everyday actions require coordinated eye-hand movements. For example, reaching for a cup on the table or playing tennis. Typically, this involves an initial rapid eye movement called a saccade, which focuses the target onto the fovea, followed by a hand movement directed to the same location, 80–100 ms later. This tight spatial and temporal coupling of the eye and hand movement is the hallmark of coordination [3]. However, despite such behavioral coupling, distinct neural pathways are thought to control the initiation and stopping of these effectors [4,5]. The neural pathways involved in generating eye and hand movements are thought to diverge in the parietal cortex after common early processing in visual areas. In the macaque brain, saccade-related signals are observed in the lateral intraparietal area and frontal and supplementary eye fields, which converge onto the superior colliculus and brain stem, which, in turn, innervate the eye muscles reviewed in [6,7,8]. Hand movement-related signals are detected in the parietal reach region, the dorsal premotor cortex and motor cortex, which project to the spinal cord and then the hand muscles reviewed in [9,10]. This suggests that distinct anatomical networks initiate eye and hand movements. However, other studies have suggested that some areas within the premotor cortex [11,12], frontal eye fields [13,14], posterior parietal cortex [15,16,17,18] (see [19,20]), and superior colliculus [21,22,23,24] that respond to both eye and hand movements. These regions possibly contribute to the neural mechanism mediating eye-hand coordination but are distributed across the brain. (reviewed in [25]). A similar issue arises when addressing how coordinated eye-hand movements are stopped. Prior research has shown that stopping eye and hand movements are thought to be mediated by anatomically separate regions. Eye movements are thought to be stopped by distinct populations of neurons in the frontal eye field and superior colliculus [26,27,28], while cortical regions, such as the right inferior frontal cortex, pre-supplementary motor area, premotor and primary motor cortex, are important for stopping manual movements [29,30,31,32]. However, Leung and Cai (2007) [33] have reported both overlapping and non-overlapping activations in the prefrontal cortex during the stopping of saccades and button presses, suggesting that both common or separate nodes might mediate the stopping of coordinated eye–hand movements. Similarly, others have suggested that the basal ganglia might play a role in the stopping of both eye [34,35] and hand [36,37] movements. While these studies are noteworthy, they fall short of providing a computational mechanism for understanding the control of eye–hand coordination. 

In this review, we outline our approach, based on behavior, to study the computational mechanism that mediates the initiation and inhibition of coordinated eye–hand movements. We tested three different versions of stochastic accumulation-to-threshold models [38,39] and identified that a common command model fits eye and hand reaction time (RT) distributions and also predicts the correlation between the RTs. To understand the mechanisms inhibiting coordinated eye–hand movements, we used a version of the double-step task [40] called the redirect task [41,42] (also called stop-change task [43]). We used behavioral measures, such as (1) compensation functions (the probability of erroneous response as a function of time—analogous to inhibition functions [44]), and (2) Target Step Reaction Time (a measure of the stopping latency—analogous to Stop Signal Reaction Time (SSRT) [45]). Additionally, we used modeling approaches based on the framework of the independent race model [46] to suggest that a unitary inhibitory process is responsible for stopping coordinated eye-hand movements. Subsequently, we observed that in certain other behavioral contexts, the eye and hand movements could be initiated and inhibited by separate commands and inhibitory signals, respectively. Thus, we suggest that the initiation of eye and hand movements operates in two modes—coupled and decoupled. The coupled mode operates when a behavioral context requires strong coupling between the two effectors (e.g., when reaching for a cup). In this case, the movements are generated by a ‘common command’. The decoupled mode operates when a behavioral context requires weaker coupling between the two effectors (e.g., when an experienced drummer plays drums). In this case, the movements are generated by separate, independent commands. More importantly, we propose that the mechanism that governs the stopping of coordinated eye–hand movements also depends on the mode that initiates the movements. In the coupled mode, when a common command initiates both movements, a unitary effector-independent inhibitory process attempts to inhibit them. Conversely, in the decoupled mode, when separate commands initiate the movements, separate effector-specific inhibitory processes attempt to inhibit them. We also propose that the brain flexibly chooses the mode depending on the behavioral context, and indeed can switch between the two modes on a trial-by-trial basis. We start by describing the coupled mode, then the decoupled mode, and finally end by suggesting how these computational mechanisms might relate to neural mechanisms. 

## 2. Computational Mechanisms That Mediate the Initiation of Eye-Hand Movements

Eye–hand movements coordinated to a peripherally presented target are characterized by three salient features: (1) the mean eye RT is typically 80–100 ms less than the mean hand RT [47,48], (2) the correlation between the eye and hand RT is high [49,50], and (3) the variability of the eye and hand RT distributions (quantified using standard deviation, SD) are similar. While the first two findings are ubiquitous in the field, the last finding has hardly been considered, and is a crucial diagnostic in unraveling the mechanism of eye–hand coordination. This is because the SD of any RT distribution under stochastic accumulation scales linearly with its mean [51,52]; in the case of coordinated eye–hand movements, this linear scaling was not observed [53]. This is diagnostic of a ‘common command’ (i.e., a single source of variability) [48,54,55,56] that underlies the initiation of coordinated eye–hand movements. 

We tested the ability of three different computational mechanisms (Figure 1) to explain the behavior when participants made coordinated saccade and reaching movements toward a target [53,57]. The motor plan for each effector was modeled as a stochastic rise-to-threshold accumulator where the time of threshold crossing represented the RT in a trial. The simplest mechanism of coordination was the **independent model** in which eye and hand movements are generated by independent systems operating in parallel, driven by a common visual target (Figure 1A). This passive model could not predict any of the three salient features of the eye–hand behavior—mean and SD of eye and hand RT distributions and high RT correlation. We also tested a mechanism called an **interactive model**, which incorporated an active mechanism of coordination (Figure 1B). This was based on the experimental observation that RTs were different between the coordinated and alone conditions. In the alone condition, the eye and hand movements were executed separately. Notably, eye RTs were delayed by 50 ms, while the hand RTs were faster by 100 ms in the coordinated condition compared to the alone conditions [53]. We incorporated this RT modulation in our model as interactions—an inhibitory interaction from the hand that delays the saccade onset and an excitatory interaction from the eye that speeds up the hand movement during coordinated movements. Despite incorporating an active mechanism of coordination, this model failed to predict two of the salient features of eye–hand behavior, i.e., the high RT correlations and the similarity in SDs of eye and hand RT distributions. A **common command model** was also tested (Figure 1C). This model suggests that eye and hand effectors are controlled by a common command, modeled as a single stochastic accumulator. When this accumulator crosses the threshold, the eye movement is initiated and is followed by a hand movement after a ‘delay’, which represents the time taken to initiate the hand movement. This mechanism could account for all three salient features—the difference in the means of eye and hand RTs, the similarity of SDs of eye and RT distributions, and also the high RT correlations. Furthermore, we validated this model by electromyography (EMG) of the arm deltoid muscles, which showed that muscle activation preceded saccade onset by 50 ms and was strongly correlated with saccade onset. More importantly, the biomechanical delay measured from EMG onset to hand onset was strongly correlated with the hand delay estimated from the common command model for each subject. These evidences suggest that the common command model is a biologically plausible mechanism that generates coordinated eye–hand movements [53].

## 3. Control of Coordinated Eye–Hand Movements

Current evidence regarding the control of eye and hand movements is largely derived from studies that used the countermanding task to assess eye and hand SSRT when these movements had to be inhibited in isolation. These studies showed that it takes longer to inhibit a hand movement, ~200 ms [46,58,59], than an eye movement, ~100 ms [26,60,61]. Similar results have been obtained by other studies involving coordinated eye–hand movements, indicating that the ability to stop eye movements is different from the ability to stop hand movements [62,63,64]. These latter studies, however, have not considered the mechanism that initiates eye–hand movements. This may be critical since the nature of control employed to stop these movements may vary depending on the mechanism of coordination used to initiate them. Following from the above, if coordinated eye–hand movements employ a common dedicated circuit to initiate, then it stands to reason that the stopping of such movements may also entail a common inhibitory mechanism. We tested this reasoning in our subsequent experiments, using the redirect task [65]. 

The redirect task had two types of trials—no step trials (60% of trials) in which the subject made eye and hand movements to the peripheral target and step trials (40% of trials). In the step trials, after a delay called the target step delay (TSD), a second target appeared at a different location. In these trials, participants tried to withhold their response to the first target and then redirect their response to the second target. Thus, in both countermanding and redirect tasks, participants try to stop their response to the first target in a minority of trials. Additionally, in the redirect task, they have to redirect their response to the second target. The behavioral responses from redirect tasks parallel those seen in countermanding tasks. (1) With increasing TSD, erroneous responses (movements made to the first target, i.e., unsuccessful redirection) increase. This produces a psychometric function, called a compensation function, which measures the probability of an erroneous response as a function of TSD (analogous to the inhibition function in countermanding tasks). (2) Redirected movements are also approximated by race models [42,57,60]. According to the general race model formulation, whether or not a prepotent response is stopped depends on the outcome of a race-to-threshold between a GO process that initiates the movement, and a STOP process that inhibits it. If the GO process reaches the threshold first, then the movement is initiated; however, if the STOP process reaches the threshold first, then the movement is inhibited. Thus, using behavior and modeling, we were able to establish the computational mechanisms that mediate the stopping of coordinated eye–hand movements in different behavioral contexts. (3) The race model can be used to estimate the Target Step Reaction Time (TSRT), the average latency to redirect movements (analogous to the Stop Signal Reaction Time in countermanding tasks).

In our study [65], participants performed the redirect task in three different conditions: (1) the eye-alone condition, where only saccades were made, (2) the hand-alone condition, where only reaches were made, and (3) the eye-hand condition, where both saccades and reaches were made. This allowed us to compare how the stopping performance changed depending on whether the effector was executed by itself (alone condition) or along with the other effector (coordinated condition). In the alone condition, the TSRT for the eye was significantly less than that of the hand (Figure 2F, left). Additionally, the compensation functions for the eye and hand were distinct from each other (Figure 2B). Taken together, this suggests that in the alone condition, the two effectors are inhibited by separate, effector-specific STOP processes. However, when we compared the behavior in the coordinated condition, we observed an interesting result: the eye TSRT was less than that of the hand (Figure 2F, middle), but the eye and hand compensation functions were comparable to each other (Figure 2D). 

To understand this conundrum of similar compensation functions despite dissimilar TSRTs, we considered the effect that RT distributions have on the compensation functions [65]. Theoretically, the nature of the compensation function depends on the outcome of the race between the GO and STOP processes. Thus, if the GO process, characterized by the no-step RT distribution, is slower, then it is expected to shift the compensation function to the right. We tested this expectation across the three conditions. In the alone condition, the difference between the means of the eye and hand RT distributions (Figure 2A) was comparable to the differences between the means of their compensation functions (Figure 2B). This suggests that the difference seen in the compensation functions during the isolated execution of the effectors is entirely driven by the RT differences (symmetric shift Figure 2E, left). However, in the coordinated condition, the difference between the means of the eye and hand RT (~100 ms; Figure 2C) was significantly higher than the difference between the means of their compensation functions (~35 ms; Figure 2D). This peculiar shift of the no-step RT, without affecting the compensation function (i.e., non-symmetric shift), was specific to the coordinated condition (Figure 2E, right). This suggests that the nature of control during the stopping of eye and hand movements is distinct between the alone and coordinated conditions. Previous studies, which have reported such changes in RT without corresponding changes in compensation functions, have attributed it to a ballistic stage that is immune to inhibitory control and reflects a ‘point of no return’ during movement initiation [66,67]. We, therefore, tested the presence of a ballistic stage during the control of coordinated eye–hand movements [65].

## 4. A Ballistic Stage Explains Redirect Behavior for Coordinated Eye–Hand Movements

In our study [65], we estimated the ballistic stage, using numerical simulations, which involved fitting the observed compensation function of the hand during coordinated eye-hand movements. Using this method, we estimated the last 45 ms of the hand motor preparation to be ballistic in nature. Since the electromechanical delay or the time between the EMG onset and hand movement is ~100 ms [55,68], we concluded that the ‘point of no return’ for the hand movement may appear after the EMG onset. We reasoned that the ballistic stage results from the interplay between the common GO process that initiates eye–hand movements and the unitary STOP process (Figure 3). When the GO process wins the race, resulting in an eye movement, the inhibitory process decays or is actively inhibited. This absence of an active inhibitory process results in the hand movement following the eye to the erroneous target. As a consequence, the compensation functions of the eye and hand are comparable despite significant differences in their RTs in the coordinated condition. Recent studies have also provided additional evidence in favor of the existence of a ballistic stage in manual responses. Muscle activity decreases ~60 ms prior to the behavioral measure of the stopping latency (SSRT), suggesting that this time interval may reflect a ballistic stage during which the inhibitory process cannot intervene and stop the response [69,70,71]. Such a mechanism also resolves the paradox of different TSRTs for the eye and hand in the coordinated condition. When this ballistic stage of 45 ms is incorporated, the hand TSRT in the coordinated condition becomes comparable to the eye TSRT (Figure 2F, right). The similarity in the eye and hand TSRT and the comparable eye and hand compensation functions suggests that a unitary, effector-independent inhibitory mechanism mediates the control of coordinated eye–hand movements [65].

## 5. A Unitary STOP Model: Salient Features and Validation

The idea that a unitary STOP process inhibits coordinated eye–hand movements contradicts previous studies that have suggested distinct inhibitory processes for eye and hand movements [62,63]. To validate the unitary STOP model, we simulated race models with a unitary STOP process and a ballistic stage in the hand motor plan [65]. This model was able to account for all the observed behavioral measures, such as compensation functions and proportions of correct and incorrect trials. The performance of the unitary STOP model was then compared to the performance of an alternate model which postulated that separate, effector-specific STOP processes inhibited eye and hand movements. However, both models were able to predict all features of the eye–hand redirect behavior to the same extent. Using a statistical model selection criterion (Akaike Information Criterion), the unitary STOP model better predicted redirect behavior with a smaller number of free parameters [65].

Further evidence beyond this statistical criterion also validated the existence of a unitary STOP process for controlling coordinated eye–hand movements. As mentioned above, the GO process initiating the coordinated eye–hand movements was modeled as a common accumulator because the variability in the eye and hand RTs were comparable. Similarly, a unitary STOP model predicts that the variability in the redirect behavior of the eye and hand should be comparable. To test this [65], we estimated the variability of the STOP process by replotting the compensation functions, using a ZRFT (*z*-score relative finishing time) normalization process [46]. Indeed, the STOP variability was comparable between the eye and hand in the coordinated condition, validating the predictions of the unitary STOP model. In contrast, when the movements were executed in isolation, the STOP variability of the eye and hand were significantly different. The latter is consistent with our previous results that suggested that separate, effector-specific STOP processes are responsible for inhibiting eye and hand movements executed in isolation. 

To test the unitary STOP mechanism experimentally, a variant of the redirect task (selective redirect) was used [65]. Participants were asked to perform a coordinated eye-hand movement in 60% of the no-step trials. During the remaining 40% of trials, in separate blocks, they were instructed to inhibit the eye (Eye-Stop); inhibit the hand (Hand-Stop); or inhibit both effectors (Both-Stop) (Figure 4). We hypothesized that if a unitary STOP process is engaged to inhibit coordinated eye–hand movements, participants would not be able to selectively stop one of the two effectors. Consistent with this idea, the hand was inhibited in the Eye-Stop condition and the eye was inhibited in the Hand-Stop condition even though the behavioral context did not require it. In other words, the redirect performance across the three conditions was not significantly different for the eye (Figure 4A,C) and hand (Figure 4B,D) effectors. This experiment showed strong behavioral evidence that eye and hand effectors, when coordinated, are controlled by a unitary STOP process that inhibits both effectors [65].

Finally, all the evidences observed in the coordinated condition, including (1) similarity of compensation functions, (2) similarity of the eye and hand TSRT after incorporating a ballistic stage, (3) superiority of the unitary STOP model over separate STOPs model as assessed by Akaike Information Criterion, (4) similarity in STOP variability, and (5) participants’ inability to selectively stop one effector in the selective redirect task, suggest that a unitary, effector-independent STOP process is responsible for inhibiting coordinated movements when they are initiated by a common command.

## 6. Computational Mechanism Mediating Flexible Initiation of Coordinated Eye–Hand Movements 

The majority of our day-to-day eye and hand movements are temporally coupled. However, in certain behavioral contexts, it might be advantageous to not couple the two movements. Consistent with this, studies have reported heterogeneous levels of temporal coupling between the two effectors. Some studies have reported weak correlations ranging between 0.1 and 0.4 [47,72,73,74], while others have reported moderate to high correlations ranging between 0.6 and 0.9 [50,53,75,76]. These suggest that flexible coupling between the two effectors is based on the task context. While the common command model can account for the tight coupling between the effectors that is seen in certain behavioral contexts, it cannot account for the varying low levels of coupling seen in other behavioral contexts. 

We studied the computational mechanism that allows such flexible coupling [77], reviewed in [78]. We hypothesized that coordinated eye–hand movements operate in two modes—coupled and decoupled. Each of these modes of coordination has specific behavioral signatures. In the coupled mode, a single GO process is responsible for initiating both movements; hence, there is a single source of variability underlying both effectors. This predicts high eye–hand RT correlation and that the SD of the two RT distributions will be similar. Conversely, in the decoupled mode, separate GO processes are responsible for initiating the two movements; hence, there are distinct sources of variability underlying the effectors. This predicts low eye–hand RT correlations and that the SD of the RT distributions will be dissimilar. We tested this hypothesis by asking participants to perform two tasks specifically designed to induce them to operate in either the coupled or decoupled mode [77].

Participants performed Go tasks in two contexts—a search task (coupled-mode; Figure 5A) and dual task (decoupled mode; Figure 5B). We reasoned that the initiation of the two effectors would be coupled or decoupled depending on whether the Go cues for the two movements were common or not, respectively. In every trial of the search task, participants had to make eye–hand movements to a common target presented among distractors, i.e., the Go cue for both effectors was common. In contrast, in the dual task, the Go cue for the eye was the appearance of a peripheral target, while the Go cue for the hand was a tone. Further, the tone was presented only in a minority of trials, ensuring that the target appearance was not predictive of the Go cue for the hand. We hypothesized that the behavior in the search task would be consistent with the prediction of a common command model (similar SDs of eye and hand RT distributions and high RT correlation), while the behavior in the dual task would be consistent with the predictions of the separate command model (dissimilar SDs of the eye and hand RT distributions and low RT correlation). The results were consistent with this prediction (Figure 5 table). In the search task, RT correlations were high (~0.8) and the SD of eye and hand RT distributions were not significantly different from each other despite a ~90 ms difference in their means. Further, this behavior was fit well by the common command model but not by the separate commands model. Conversely, in the dual task, RT correlations were low (~0.3) and both the mean and SD of the eye RT distribution were significantly less than the mean and SD of the hand RT distribution, respectively. Further, the behavior in the dual task could be explained by the separate commands model but not by the common command model. Thus, these results suggest that there are two mechanisms mediating the initiation of eye-hand movements and that, depending on the task context, the brain is biased to one of the mechanisms [77].

## 7. Stopping of Flexibly Initiated Eye–Hand Movements

As mentioned above, there is debate as to whether the stopping of eye–hand movements is mediated by unitary or separate STOP processes [62,63,64,65]. These heterogeneous results might be because the movements were generated in task contexts, which biased the brain toward a coupled or decoupled mode. Thus, we hypothesized that, depending on the mode in which coordinated eye–hand movements are generated, the brain can flexibly employ unitary or separate STOP processes to inhibit the movements. To test this, we added a redirect component to the coupled and decoupled Go tasks to study the redirect behavior in these contexts [79]. The extension of the search task was called the search redirect task (Figure 6A). Here, 60% trials were no-step where eye–hand movements had to be made to the target among distractors, while 40% were step trials where the target jumped to another location after a target step delay (TSD). In these trials, movements had to be made to the second target location and not the first target location. Thus, the search redirect task represented the task context where, presumably, there was a common command and unitary STOP at work. Conversely, the extension of the dual-task was the dual redirect task (Figure 6B), again with 60% no-step and 40% step trials. Thus, the dual redirect task represented the task context where presumably there were separate commands and separate STOPs at work.

We initially tested the no-step RT distributions and correlations and found convincing evidence that responses in the no-step search task were generated by a common command in the coupled mode, while responses in the no-step dual task were generated by separate commands in the decoupled mode [79]. Next, we compared the eye and hand compensation functions from the step trials between the two task contexts. We reasoned that in the coupled mode (search redirect task), the movements would be inhibited by a unitary STOP process, resulting in similar eye and hand compensation functions (Figure 6C). In the decoupled mode (dual redirect task), the movements would be inhibited by separate STOP processes, resulting in dissimilar compensation functions (Figure 6E). The results were consistent with this prediction (Figure 6D). We also estimated the variability of the STOP processes by replotting the compensation functions using ZRFT (*z*-score relative finishing time) normalization and observed that the variability was larger in the decoupled mode compared to the coupled mode. This, again, suggested that in the coupled and decoupled conditions, unitary STOP and separate STOP processes are responsible for inhibiting eye and hand movements, respectively [79]. 

Further support to this conclusion came from simulation results of comparing unitary and separate STOPs models. The unitary STOP (Figure 6F) but not the separate STOPs model (Figure 6G) was able to predict the behavior in the search redirect task. Conversely, the separate STOPs (Figure 6H) but not the unitary STOP (Figure 6I) model was able to explain the behavior in the dual redirect task. Taken together, this highlights the flexibility that exists in the brain where the task context biases the mechanism toward a coupled mode (common command and unitary STOP) or a decoupled mode (separate command and separate STOPs) [79]. We speculate that a higher-order executive process evaluates the behavioral context and chooses the appropriate mode of operation.

## 8. Parallels between Common and Separate Stops and Global and Selective Stopping

Thus far, we have suggested that there are two computational mechanisms that mediate the stopping of coordinated eye–hand movements—(1) a unitary STOP used in the coupled mode, and (2) separate STOPs used in the decoupled mode. How do these computational mechanisms relate to neural mechanisms? Interestingly, our conclusions largely echo the research on the notion of distinct modes of stopping, namely, global vs. selective stopping, which determines whether stopping has a broad impact on motor cortical areas or is selective to the motor area that needs to be inhibited.

Global stopping is thought to be recruited when there is a rapid reactive need to stop a movement and is thought to be implemented by rapid inhibition of all motor cortical areas. This rapid stopping is thought to be mediated by the hyper-direct pathway that connects prefrontal cortical areas to the subthalamic nucleus (STN) of the basal ganglia [80,81,82]. This activation of the STN via the output of the basal ganglia rapidly cuts off the thalamocortical drive, leading to global inhibition of the motor cortex within a few hundreds of milliseconds [70,83]. This global inhibition of the motor cortex can be measured using transcranial magnetic stimulation (TMS). Numerous TMS studies have demonstrated that in trials where a response is successfully stopped, there is a decrease in TMS evoked muscle responses (motor evoked potential, MEP) in those muscles, which are not involved in the task, e.g., decreased MEP of the leg when stopping the hand [84], decreased MEP of the hand when stopping the eye [85], decreased MEP of the hand when stopping speech [86], and decreased MEP of the left hand when stopping the right hand [87]. 

In contrast, selective stopping is thought to be recruited when there is a need to selectively stop the effector without affecting other effectors. Such stopping is slower and more effortful. This stopping is thought to be mediated by the slower indirect pathway of the basal ganglia [88], and in this case, decreased MEP is seen only in the muscle that has to be inhibited [84,89] (see [71]). These neural mechanisms fit our conclusions derived from behavioral studies. Taken together, we propose that the control of coordinated eye–hand movement that is mediated by a computational unitary STOP process is neurally implemented by a global stopping mechanism while the control of coordinated eye–hand movements that are mediated by computationally separate STOP processes is implemented by a selective stopping mechanism. These predictions can be tested by future studies. 

## 9. Significance

Our proposed framework suggests that there are two modes in which coordinated eye-hand movements operate: (1) a coupled mode in which both effectors are initiated and inhibited by a common command and a unitary inhibitory process, and (2) a decoupled mode in which effectors are initiated and inhibited by separate commands and separate inhibitory processes. This may be ecologically and behaviorally relevant as it would provide animals with the flexibility required to interact with and survive in natural environments. Animals in their natural environment deal with situations that can be time-sensitive and require rapid responses or those that are time-insensitive for responses. For example, consider an arboreal primate jumping from one tree branch to another. The ability to quickly generate precise, temporally coupled eye–hand movements are crucial for survival in this case. When the same animal encounters a time-insensitive situation, e.g., cracking a nut, then more adaptable slow movements may be warranted. Having a decoupled mode in which eye and hand movements are separately controlled may allow the animal the flexibility to deal with the latter situations. In a similar vein, a unitary, effector-independent stopping mechanism may provide an animal behavioral advantage during ecologically relevant, highly time-sensitive behavioral situations, such as evading a predator/attacking a prey. It may help the animal to rapidly stop its action and change course in response to changing environmental conditions. The effector-dependent stopping mechanisms are slower but may provide flexibility during time-insensitive behaviors. Such flexibility in brain and behavioral mechanisms have also been observed in selectively stopping actions [30,90] (see [91]). 

Besides rapidly stopping due to sudden changes in the environment (reactive stopping), we can sometimes prepare to stop (proactive stopping), such as preparing to stop in anticipation of the traffic light turning red. Reactive and proactive stopping are thought to require separable neural and cognitive processes [88,92]. Based on this distinction, we note that our studies tested reactive stopping of coordinated eye–hand movements. In our framework, higher-order executive processes determine the mode in which the movements are reactively stopped, i.e., by a unitary STOP or separate STOPs. How does our proposed framework relate to proactive stopping? Proactive stopping may require higher-order executive processes that adapt motor strategies for optimizing stopping behavior. We speculate that such executive processes might interact with the executive processes that determine the mode of eye–hand control. Consider the situation where one proactively knows an effector needs to be selectively stopped many times (i.e., far more than our study, Figure 4). Here, with time and practice, the initiation of movements might become decoupled so that selective stopping becomes more efficient. For example, imagine someone learning to play drums. Initially, they find it hard to decouple the effectors, i.e., selectively initiate and inhibit one effector, even if they know the task requirements in advance. However, with time and practice, proactive control processes may shape behavior such that the effectors get decoupled, allowing more efficient and selective control. Further studies are required to test this speculation.

## 10. Conclusions

Using a combination of behavior, computational modeling, and electromyography, we propose a computational mechanism that mediates the initiation and inhibition of coordinated eye–hand movements. We suggest that, depending on the task context, eye–hand movements may be initiated and inhibited in one of two modes: a coupled mode where a common command initiates and a unitary STOP inhibits both movements; and a decoupled mode where separate commands initiate and separate STOPs inhibit the two movements. Such a mechanism can explain eye–hand behavior in numerous contexts and reconciles the heterogenous results reported by previous studies. We hope that the computational framework that we have developed and tested rigorously will aid cognitive neuroscientists in discerning the neural mechanism of eye–hand initiation and inhibition.

## Figures and Tables

**Figure 1 brainsci-11-00607-f001:**
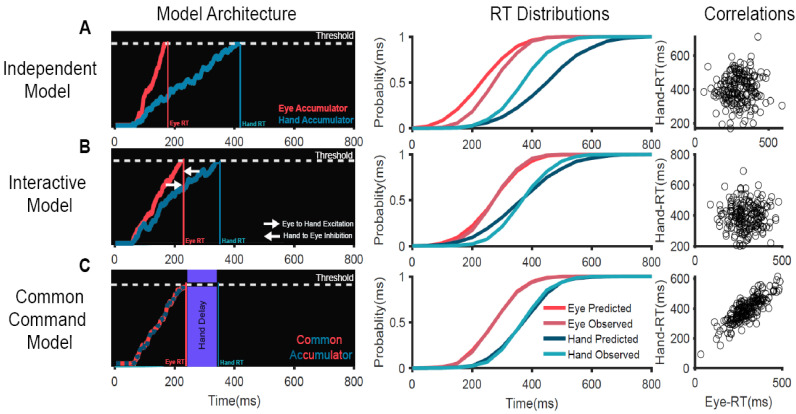
Comparison of different computational mechanisms mediating the initiation of coordinated eye–hand movements [53]. (**A**) An independent model in which eye and hand effectors are controlled by independent accumulators driven by common visual inputs. (**B**) An interactive model in which the eye and hand accumulators interact with each other—the eye provides excitatory inputs to the hand while the hand provides inhibitory inputs to the eye. (**C**) A common command model in which eye and hand movements are controlled by a common accumulator. When this accumulator crosses the threshold, the eye movement is initiated and is followed by the hand movement after a peripheral ‘delay’. Left panels: Schematic representation of the mechanisms of eye–hand coordination. Eye (red) and hand (dark blue) motor plans are represented as stochastic accumulators that start after a visual delay of 60 ms after the target onset (time = 0 ms). The accumulators rise to a threshold represented by the white dotted line. The time at which each accumulator crosses the threshold represents the RT in that trial (eye RT, red vertical line; hand RT, dark blue vertical line) (middle panels). Comparison of the observed RT cumulative distribution to that predicted by the model on the left (predicted eye, dark red; predicted hand, dark blue; observed eye, light red; observed hand, light blue) for a representative subject. Right panels: Scatter plot of the predicted eye (x-axis) and hand (y-axis) RT, generated by the model on the left. Each dot represents a simulated trial.

**Figure 2 brainsci-11-00607-f002:**
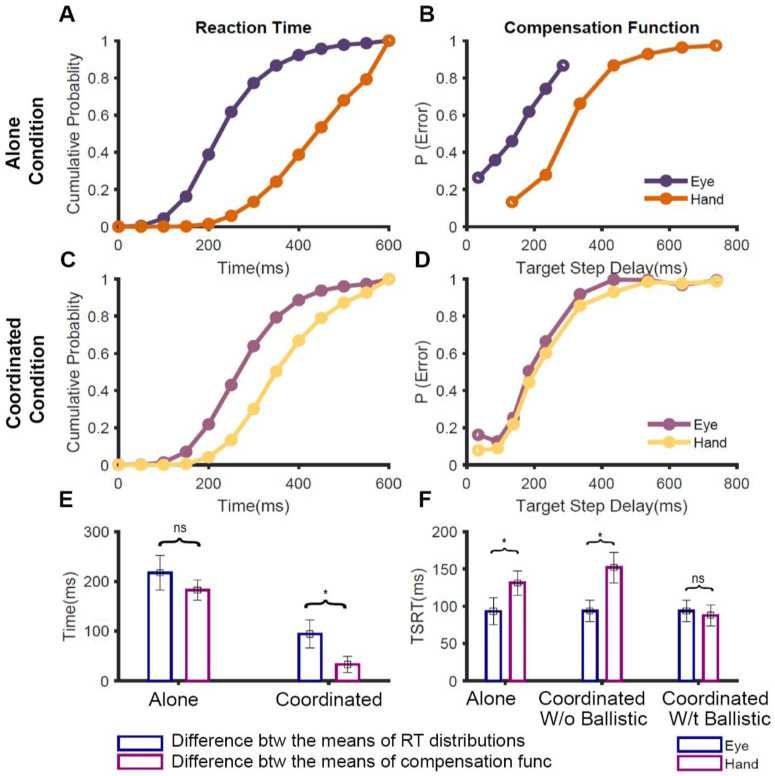
Comparison of redirect behavior between alone and coordinated conditions [65]. (**A**) The average cumulative RT distributions of eye (violet) and hand (orange) across participants in no-step trials when the movements were executed in isolation. (**B**) The average compensation functions of eye (violet) and hand (orange) across participants when they were redirected in the alone conditions. (**C**) The average cumulative RT distributions of eye (purple) and hand (yellow) across participants in no-step trials when the movements were executed in the coordinated condition. (**D**) The average compensation functions of eye (purple) and hand (yellow) across participants when they were redirected in the alone conditions. (**E**) A bar graph showing the symmetrical shifts in the alone condition, i.e., differences seen in the RT is fully mirrored in the compensation functions, and non-symmetrical shifts in the coordinated condition, i.e., differences in RT only partially mirrored in the compensation functions. The blue bar shows the average difference between the mean RTs of eye and hand while the magenta bar shows the average difference between the means of eye and hand compensation functions. (**F**) A bar graph showing the TSRT calculated for eye (blue) and hand (magenta) during the alone condition and coordinated condition, with and without incorporating a ballistic stage * *p* < 0.05, ^ns^
*p* > 0.05.

**Figure 3 brainsci-11-00607-f003:**
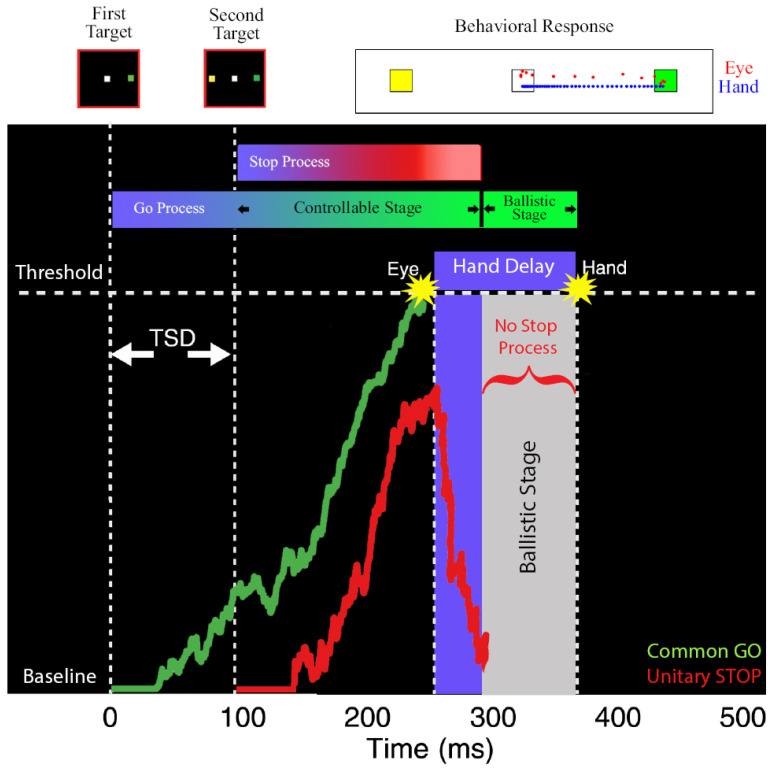
Schematic depicting the ballistic stage for coordinated eye-hand movements [65]. The GO process (green) is activated after a visual delay following the first target onset and the STOP process (red) is following the second target onset. These two processes race to a threshold. The eye movement is initiated when the GO process crosses the threshold. After the GO process wins the race, the STOP process is actively inhibited or decays. The lack of an active inhibitory process results in a ballistic stage (grey) leading to the inevitable execution of hand movements. Top: Eye (red dots) and hand (blue dots) movement traces from an error trial where both effectors landed on the first target (green) and not the second target (yellow).

**Figure 4 brainsci-11-00607-f004:**
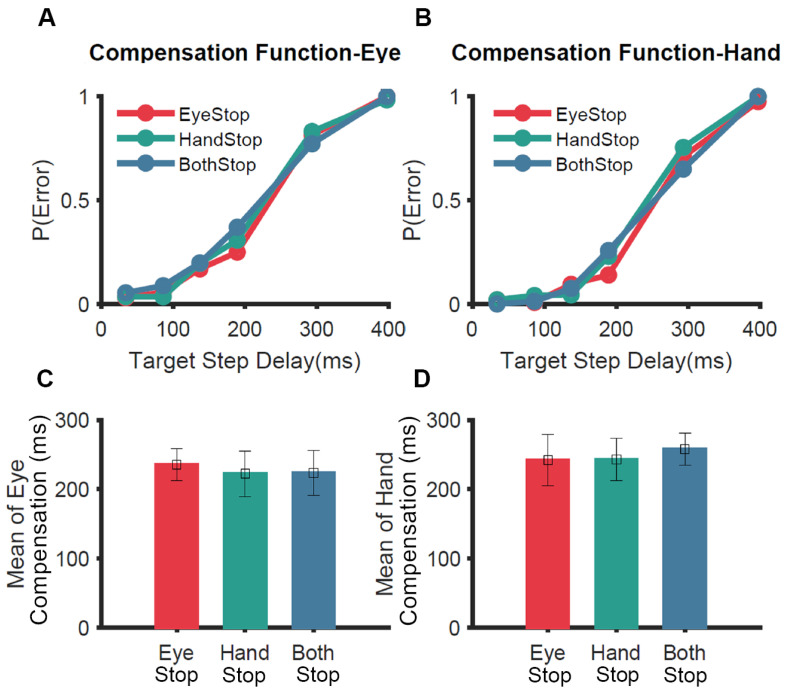
Comparison of selective redirect behavior [65]. In a selective redirect task, participants performed a coordinated eye–hand movement to peripheral targets on 60% of trials. In 40% of the trials, in separate blocks, they were instructed to inhibit the eye (Eye-Stop, red); inhibit the hand (Hand-Stop, green); or inhibit both effectors (Both-Stop, blue). (**A**) The average compensation function of the eye across participants. (**B**) Same as A but for the hand. (**C**) Comparison of the means of compensation functions of the eye for Eye-Stop (red), Hand-Stop (green) and Both-Stop (blue) trials. (**D**) Same as C but for the hand.

**Figure 5 brainsci-11-00607-f005:**
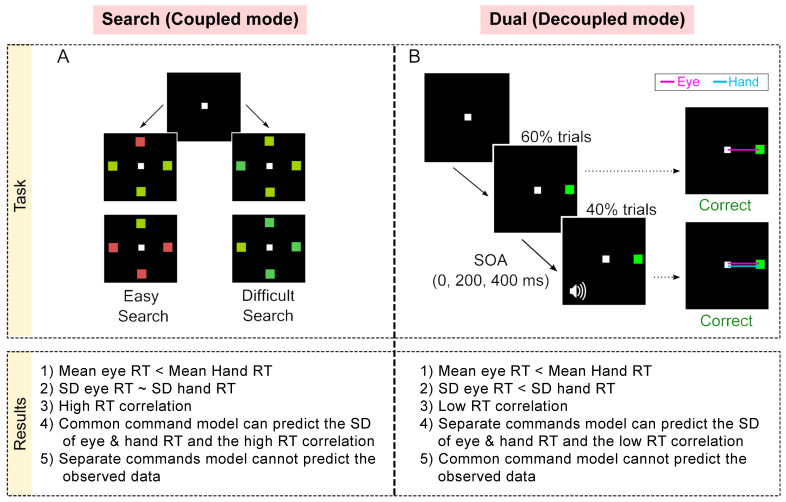
Summary of tasks and observed behavior [77]. (**A**) Search task: In each trial, participants had to move to the target among distractors. The search could be either easy (red among green or vice versa) or could be difficult (blue-green among green and vice versa). (**B**) Dual task: In 60% of trials, participants had to make an eye movement to the green target presented in the periphery (Eye Alone). In 40% of the trials, after a Stimulus Onset Asynchrony (SOA) of 0, 200, and 400 ms per tone was presented, which indicated that participants had to make a hand movement to the target as well. The comparison presented here shows only the SOA 0 ms trials (coordinated), i.e., the Go cues for both effectors were presented at the same time but note that the Go cues were visual and auditory for the eye and hand, respectively. Bottom: Table comparing the behavioral and simulation results.

**Figure 6 brainsci-11-00607-f006:**
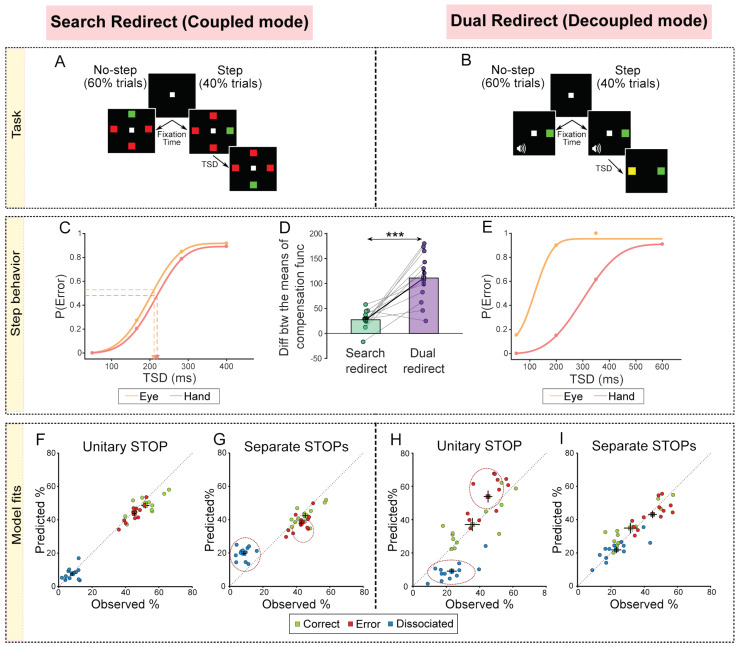
Summary of tasks, observed behavior and simulation results [79]. (**A**) Search redirect task: 60% of trials were no-step where the target did not shift, and 40% of trials were step trials where the target shifted. In each no-step trial, participants had to make movements to the odd colored target (red among green or green among red). In the step trials, after a delay called the Target Step Delay (TSD), the target shifted to a new location and the participants were now expected to move to the second target location instead. (**B**) Dual redirect task: 60% of trials were no-step where participants had to make an eye movement to the peripheral target and also make a hand movement to it if a tone was also presented. A total of 40% of the trials were step trials where a yellow target was subsequently presented after a TSD. In these trials, participants had to make an eye and hand movement to the yellow target. (**C**) Cumulative weibull fits for an exemplar participant for the eye (orange), and hand (red) in the Search redirect task (data, dots). The arrow indicates the means of the compensation functions, i.e., the TSD corresponding to the mean of the error (unsuccessful redirection). (**D**) Same as C but in the Dual redirect task. (**E**) Comparison of the difference between the means of the eye and hand compensation functions between the Search redirect (green) and Dual redirect (purple) tasks. Each dot represents a participant, while the bar and cross hair represent the mean ± s.e.m. across the population. (**F**) Comparison of the trial percentages behaviorally observed and that derived from the simulations of the unitary STOP model for the Search redirect task. The green dots represent the correct trials (both eye and hand movements to the second target), the red represents the error trials (both eye and hand movements to the first target), and the blue represents the dissociated trials (eye and hand movements to different locations). (**G**) Same as F but for the Separate STOPs model. Dotted circles indicate that predicted % does not match the observed %. (**H**) Same as I but for the Dual redirect task. (**I**) Same as G but for the Dual redirect task. *** *p* < 0.001.

## Data Availability

The datasets generated and analyzed as part of the original studies are available from the corresponding authors on reasonable request.

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
