# Peer review of "Computational Mechanisms Mediating Inhibitory Control of Coordinated Eye-Hand Movements"

_brainsci, 2021, doi:10.3390/brainsci11050607_

Round 1

Reviewer 1 Report

This manuscript presents a comprehensive review of past studies by the authors about inhibitory control of eye-hand coordinated movements. The studies are presented in a unified and comprehensive perspective explaining the rationale that led to each single research question. I was not familiar with the single studies but this manuscript provides a good overview clearly pinpointing the evidence substantiating the proposal, put forward here, of a system for the initiation and stopping of eye and hand movements and flexibly operating either in a coupled or decoupled mode, depending on the behavioral context.

The use of cognitive modeling to make predictions about behavioral results linking them to possible neural mechanisms and thus verifying theoretical hypotheses is in my view commendable. Still, there are some points to be addressed to make the manuscript even more compelling:

  • Since the title, the focus is on a computational architecture, yet I'm not sure what architecture would that be: there is no computational architecture presented with multiple modules or layers through which information flows.  I'd rather talk of a computational model of neural mechanisms which makes predictions about and explains behavioral results. There is of course a neural architecture generating those observations, yet the used race models seem rather agnostic as to which module/area does what, rather modeling the underlying computational mechanism.
  • In sections 2-7, different studies are reviewed. I had really a hard time retrieving which study was presented in each section since often referred along with related work or at the wrong place. The studies should further be mentioned by their bibliographic reference in the caption of the figures which depict the corresponding results. In section 3, for example, on lines 179 and following an experimental paradigm of a study is presented but no reference to the related publication is given. Line 210 starts with "In our study,..." and it's not clear which one (a few lines above 3 were cited).
  • In Figure 2E the caption mentions symmetrical and asimmetrical shifts but it should be made clearer what is the symmetry here (it can be deduced in the text but here it's a bit obscure, plus it's not clear what WM Diff stands for). 
  • On lines 220 it is stated " the eye TSRT was less than that of the hand", yet in Figure 2F for the coordinated condition it looks like the eye TSRT is slightly larger than the hand...am I missing something here?
  • Section 4 starts with "Ïn our study,..." and that is where reference 56 should be put, otherwise it seems a publication on numerical simulations is referred.
  • Again, on line 256, "Recent studies" are mentioned but cited only at the end of the following sentence. Consider moving the reference earlier for clarity.
  • I suppose Figure 3 should be cited in Section 4 too, or at least where it is appropriate in the text.
  • Again, in section 5, what study it is talked about here (lines 277-286)? Is it the same presented in lines 300-310? The reader should be allowed to get further details in the original studies, if interested.
  • In Section 6, line 331, the corresponding study is instead cited twice.
  • In Figure 4, both lines 5 in the comparison table seem to end abruptly: was the last line cut off?
  • I appreciated the discussion in Section 8 with the comparison to global and selective stopping. It would also be nice to put the presented results and the proposed mechanisms in a more evolutive and cognitive perspective: what ecological need is satisfied or for which purpose a flexible unitary system for eye-hand coordination control would give an evolutionary advantage? Is there any even speculative hypothesis that can be elaborated on here?

Reviewer 2 Report

I enjoyed this manuscript. It is well-written and will be a nice contribution to the literature.

I have several recommendations to improve the article.

1- The article focuses on initiation and termination of eye and hand in coupled and decoupled frameworks. I find the organization a bit hard to parse. The authors include a decent overview at the end of the introduction but it appears incomplete to me. A stronger recapitulation of the entire paper would be stronger with inclusion of all elements (following various sections).

2- control of eye-hand movements (section 3) focuses on countermanding. This is a needed step but there are definitely more REFs that are relevant and that have added to the field. E.g., Tarkesh Singh has articles on EHC in stroke with interesting findings. JR Rizzo et al. have interesting articles on EHC in stroke as well. Viewing EHC through the lens of pathology is very helpful, particularly regarding mechanism. Aside from the stroke circuitry work, additional REFs from Roger Carpenter are warranted.

3-Description of task (re-direct task). Although I do agree that there is novelty in the instructions given (which can be highlighted as a re-direct), this task is a double-step task. This is a well-known paradigm in the field and one that should be mentioned and used as an 'anchor'. Additionally, when 'steps' are referred to, pls leverage first or second 'step' as opposed to step. This would confuse a novice reader. Appropriate REFs are warranted here.

4- Lines 300-310 that describe the unitary stop experiment is clever. This seems to be a bit underplayed. This is more unique and can be featured as a paradigmatic highlight.

5- In general, I believe care should be taken in describing coupled and decoupled activities throughout the manuscript. In the beginning (end of intro) there is an example of looking and grasping a glass of water versus playing the drums. I am not sure I agree with the authors on use of the drum playing. A novice drum player may couple eye and hand earlier in play and continue to do so until they understand the spatial layout of their drum set. But I am quite positive that if you altered the spatial design of drum sets in specific ways, even for experts, they, too, would require more coupled activities, at least in the beginning.

6- Line 467 in closing the penultimate section, you state twice coordinated eye-hand movements. I think you mean coupled and then decoupled. 
